# Advances in the Deformation and Failure of Concrete Pavement under Coupling Action of Moisture, Temperature, and Wheel Load

**DOI:** 10.3390/ma13235530

**Published:** 2020-12-04

**Authors:** Wanguo Dong, Chunlin Liu, Xueben Bao, Tengfei Xiang, Depeng Chen

**Affiliations:** 1School of Management Science and Engineering, Anhui University of Technology, Ma’anshan 243032, China; wanguo88@ahut.edu.cn; 2School of Architectural and Civil Engineering, Anhui University of Technology, Ma’anshan 243032, China; chunlinliu@ahut.edu.cn (C.L.); xbb2020123@163.com (X.B.); xiangtf@ahut.edu.cn (T.X.); 3Institute of Green Building Materials, Anhui University of Technology, Ma’anshan 243032, China

**Keywords:** hygro-thermo-mechanical, coupling action, pavement concrete, deformation and failure

## Abstract

The deformation and cracking of concrete will lead to various deterioration processes, which will greatly reduce the durability and service life of the concrete pavement. The relating previous studies and analysis revealed that the coupling action of environmental temperature, moisture, and wheel load will cause cracking and seriously affect the normal service and durability of pavement concrete. This paper presents theoretical and numerical state-of-the-art information in the field of deformation and failure of pavement concrete under coupling action of moisture, temperature, and wheel load and draws some conclusions. (a) Concrete is a typical porous material, moisture and heat transfer theory has obtained enough data to simulate the hygro-thermo properties of concrete, and the relationship between moisture and heat is very clear. (b) There are few studies on concrete pavement or airport pavement considering the coupling action of moisture, temperature, and wheel load. (c) Concrete pavement is subjected to hygro-thermal-mechanical coupling action in service, which has the characteristics of a similar period and its possible fatigue effect. (d) COMSOL software has certain advantages for solving the coupled hygro-thermal-mechanical of concrete.

## 1. Introduction

Cement concrete pavement is widely used in the construction of the airport runway and high-grade highway in China. More than 80% of the airport pavement in China adopts cement concrete [1]. For pavement concrete, the exposed surface formed after the construction of the whole cast-in-place pavement is large, and the trend of early shrinkage cracks caused by concrete water loss is more obvious. Temperature cracks, including temperature shrinkage and temperature fatigue cracks caused by the temperature difference between day and night, are also cracks of the airport or highway pavement form. At present, there are more researches on temperature cracks [2,3,4]. It is found that the cement concrete airport and highway pavement which has not reached the design service life has been found to have various degrees of defects, cracks, and even complete damage. This phenomenon brings huge maintenance pressure and economic burden. Dry shrinkage cracks and warpage cracks are also the main forms of concrete cracks on airport pavement, especially in areas with drought, wind, less rainfall, and large evaporation, the drying shrinkage deformation of concrete is more significant. For the airport pavement during the service period, the diurnal (seasonal) temperature difference and the diurnal (seasonal) coupled with dry-wet changes present quasi-periodic changes, and the effects of temperature and humidity changes also have coupling effects. In addition, airport pavement or highway pavement also bears the action of aircraft or driving wheel load and its dynamic effect, which will make pavement concrete appear fatigue failure. Therefore, the deformation, cracking, and fracture failure of airport and highway pavement are the comprehensive embodiment of temperature, humidity fatigue effect, and wheel load fatigue effect.

This paper will start from the coupling action of moisture, temperature, and wheel load that affects the deformation and failure of concrete pavement in the service life. The basic model of the multi-physical fields of concrete, research on the fatigue effect of concrete pavement, research on the deformation and failure of concrete pavement or airport pavement (except for concrete pavement or airport pavement), and the numerical simulation method of hygro-thermal-mechanical coupling deformation of concrete are discussed. This paper will provide new research ideas for the study of concrete deformation and failure of the pavement.

## 2. The Basic Model of The Multiple Physical Fields of Concrete

Concrete is a typical porous medium material [5,6], and its internal moisture and heat transfer are similar to the transmission principle and model of general porous media materials. The mechanism of moisture transmission in concrete is the diffusion of steam and liquid water. The transmission of water vapor conforms to the principle of liquid diffusion and is also the main form of concrete wet transmission. According to the actual hole structure size of concrete, the internal water diffusion should be combined with molecular diffusion and Knudsen diffusion. These provide a theoretical basis for the numerical simulation calculation of concrete hygro-thermo coupling deformation. The heat and moisture transfer can be commonly expressed based on Fourier’s Law, Fick’s law, and the law of conservation of mass, respectively [7,8,9,10,11]. Experiments on temperature and relative humidity were conducted by Haitao Zhao [11], and a coupled model of temperature and relative humidity was proposed. The test results showed that the coupled phenomenon of temperature and relative humidity could be observed and divided into three stages.

### 2.1. Moisture Diffusion Model

Generally, it is assumed that the moisture transfer in porous cement-based materials is in the form of diffusion, and the humidity gradient is the driving force of humidity (liquid water and steam). According to Fick’s law and mass conservation equation, the one-dimensional transmission formula is as follows [12]:(1)$$\frac{\partial }{\partial x}\left(D_{m}\frac{\partial M}{\partial x}\right)+Q_{m}=\frac{\partial M}{\partial t}$$

In the formula, *M* is the moisture content: *D_m_* is the moisture diffusion coefficient. In the actual calculation, the moisture diffusion coefficient should be modified, that is, *D_mk_* = *K_f_D_m_*, *K_f_* is the Knudsen diffusion influence coefficient; *Q_m_* is the wet source.

### 2.2. Heat Transfer Model

Fourier’s law is usually used to describe heat transfer in cement-based materials. The thermal conductivity is the apparent thermal conductivity which has taken into account the influence of internal convection. The Formula (2) is as follows [12]:(2)$$\left(-\lambda \frac{\partial T}{\partial x}\right)+Q=\rho c_{p}\frac{\partial T}{\partial t}$$

In the formula, *Q* is the heat source, the heat change caused by cement hydration heat release or other non-heat transfer processes; *T* is temperature; ρ is the apparent density of cement-based materials; $c_{p}$ is the specific heat of cement-based materials; λ is the nominal thermal conductivity of cement-based materials.

### 2.3. Microprestress-Consolidation Theory

Consolidation theory [13] and micro prestressing consolidation theory [14] are widely accepted and used theoretical models. Consolidation theory provides a physical mechanism description for the basic creep and aging effect of concrete, and its B3 model [15] has been widely used in the analysis and calculation of compressive creep of mature concrete. At present, the Microprestress-Solidification (MPS) theory is widely adopted to analyze the complex influence of temperature and humidity on concrete creep. However, the MPS theory still has some shortcomings. The MPS theory was improved for concrete creep under a complex environment [16].

Under the action of uniaxial stress, the total strain of concrete can be regarded as the sum of five parts of strain, which can be expressed by Formula (3). The rheological model is shown in Figure 1.
(3)$$\varepsilon =\varepsilon^{i}+\varepsilon^{ev}+\varepsilon^{f}+\varepsilon^{sh}+\varepsilon^{T}$$

In the formula, the $\varepsilon^{i}$ is instantaneous strain; the $\varepsilon^{ev}$ is viscoelastic strain; the $\varepsilon^{f}$ is the viscous strain; $\varepsilon^{sh}$ is the shrinkage strain caused by the humidity change; $\varepsilon^{T}$ is the temperature strain caused by the temperature change.

## 3. Research on the Fatigue Effect of Concrete Pavement

For the pavement in the area with significant temperature and humidity changes, frequent takeoff and landing of aircraft or airport runway of large military multi-wheel aircraft, heavy load and frequent traffic, the pavement is more prone to deformation and damage. The problem of deformation and cracking of airport runway, highway, and urban road pavement has not been well solved because the influence factors of pavement concrete deformation and failure are not fully considered in the design stage. The above factors include temperature change, dry and wet change, and dynamic and static action of wheel load and its coupling effect. The research on the fatigue effect of airport concrete is mainly as follows: large scale airfield concrete slabs were tested by Jeffery R. Roesler [17] in the laboratory to evaluate the effect of multi-wheel gears on the fatigue resistance of concrete slabs. The test program solved the influence of peak stress ratio, stress range and stress pulse type on the fatigue resistance of concrete slabs. The mechanical properties of pavement concrete under the joint action of corrosion, fatigue, and fiber content were assessed by response surface methodology (RSM) [18]. The RSM model fitted well and indeed effectively revealed the mechanical properties of pavement concrete under the joint action of corrosion, fatigue, and fiber content. Frost damage was a common durability problem for concrete structures in cold and wet regions, and in many cases, the frost damage was coupled with fatigue loadings such as the traffic loads on bridge decks or pavements. To investigate the basic fatigue behavior of concrete materials affected by frost damage, a mesoscale approach based on Rigid Body Spring Method (RBSM) had been developed [19]. Some researchers studied the fatigue failure law of concrete pavement by adding new materials in the concrete pavement [20,21,22,23]. To sum up, concrete pavement is subjected to hygro-thermal-mechanical action in service, which has the characteristics of a similar period and its possible fatigue effect.

## 4. Research on Deformation and Failure of Concrete Pavement or Airport Pavement

In the related research on deformation and failure of airport pavement or cement concrete pavement, many researchers mainly focus on the influence of temperature, fatigue load and freeze-thaw, temperature and humidity changes on the shrinkage, and cracking of pavement concrete [24,25,26,27,28,29,30,31,32], and there is a little previous literature on the coupling of humidity, heat and wheel load or considering the effect of temperature, humidity and wheel load at the same time [33]. Yinchuan Guo [33] studied micropore deterioration and its mechanism in pavement concrete in seasonally frozen regions, the dynamic deterioration rules of micropores under different coupling levels were discussed at the micro scale. To clarify the influence of the coupling conditions on the evolution of pavement concrete micropores, a fatigue load single field and fatigue load and freeze-thaw double field were designed as the control groups. The detailed programs are shown in Figure 2 and Figure 3. Xiaolong Yang [30] studied the micropore change and its mechanism of pavement concrete in the seasonal freeze-thaw region. The coupling effect tests of fatigue load, freeze-thaw cycle, and dry wet cycle were carried out, and the evolution mechanism of micropore in seasonally frozen regions was proposed.

Seongcheol. Choi [29] et al. studied the influence of environmental temperature and humidity on the behavior of continuously reinforced concrete pavement. The drying shrinkage of concrete and the elastic modulus of coarse aggregate played a decisive role in the development of nonstructural stress of concrete pavement. The behavior analysis of continuously reinforced concrete pavement (CRCP) under environmental load was divided into two stages, namely the early stage and the late stage. Two models were proposed, and Figure 4a illustrated the structural model used to analyze the behavior of CRCP at the early stages before transverse cracking. Figure 4b showed the structural model used to analyze the behavior of CRCP at later stages after transverse crack development. Arunw. Dhawale [34] concluded that the stress in the rigid concrete slab was caused by wheel load and slab movement caused by moisture loss and temperature change in the whole slab depth. The warpage stress caused by the difference of moisture content between the top and bottom of the slab was equal to the stress caused by the heaviest load in importance, and the warpage stress response was more important than the wheel load stress. H. Tomas Yu [35] measured the response of concrete pavement surface temperature and wheel load with an instrument panel. For cracking performance, most of the pavement with stable base showed an unbonded response, although the back-calculation results showed that most of the same cross-sections an adhesive response, which may indicate that most slab cracking occurred under the corner loading. Ye. Dan and Mukhopadhyay, Anal. K [26,36] studied the warpage stress and deformation of pavement concrete slab caused by early moisture and heat transfer. It was considered that the damp heat effect had a decisive influence on pavement concrete deformation and cracking under the condition of environmental climate change, and emphasized the importance of strengthening early maintenance. Many researchers [24,25,26,27,29,37] had studied the warpage of concrete pavement, which showed that the early warpage deformation behavior of concrete pavement was not only affected by temperature change but also affected by humidity change, drying shrinkage, and temperature conditions during construction. The core viewpoints of relevant studies are shown in Table 1.

Many Chinese researchers have carried out research on warpage stress. In recent years, there have been studies considering the joint effect of temperature effect and wheel load stress. Tan Zhiming et al. [38] earlier studied the numerical simulation of temperature field, temperature warpage stress, and load stress of cement concrete pavement under different conditions. Wu Jun et al. [39,40] studied the dynamic performance test of a multi-layer pavement system under impact load and concluded that the impact resistance of rigid pavement was weak and brittle fracture occurred easily under impact load. The impact resistance performance test and numerical simulation of composite pavement structure were carried out. The three-dimensional numerical model of the composite pavement system under impact load was established. Zhao fangran et al. [41] studied the damage of the airport pavement thermal blowing-snow process and considered that the comprehensive effects of thermal shock stress, the thermal expansion force caused by ice crystal vaporization, and freeze-thaw cycle were the fundamental reasons for the damage of the airport concrete pavement. Ling Jianming et al. [42,43] used the finite element software ABAQUS to analyze the mechanical responses of cement concrete airport pavement to multiple-gear military aircraft loadings.

A 3D finite element model was established for analyzing the mechanical responses of cement concrete airport pavement to multiple-gear military aircraft loadings. The test results showed that the maximum tensile stress in the pavement occurred at the bottom of the slab, the tensile stress and the deformation of the pavement under two 8-wheeled gears were smaller than that under one 8-wheeled gear. The elasticity modulus of the slab, the resilience modulus of the base and the reaction modulus of the subgrade all had significant effects on the mechanical responses of the pavement.

Zheng Fei and Weng Xingzhong [44] analyzed the main influence factors of pavement slab stress using elastic foundation plate theory and proposed the stress calculation method of cement concrete pavement slab under aircraft load. Wang Zhenhui et al. [45] considered that the cumulative damage model of rigid pavement should consider the influence of load stress distribution of pavement slab, and established the cumulative damage optimization model suitable for rigid pavement by using the covering action curve and stress distribution function. Huang Xiaoming et al. [46] studied the temperature warpage stress of continuously reinforced cement concrete pavement and considered that the temperature warpage stress of a single slab in the Winkler foundation model can be calculated by using the warpage stress calculation formula of ordinary cement concrete slab consistent with its size. Li Xinkai et al. [47] analyzed the deformation and stress of cement pavement slab under the action of axial load and temperature. Slab deflection and stress were calculated under axial loads at different slab positions and negative or positive temperature gradient coupling. The calculated results show that the different conditions of axial loads and temperature gradient coupling will change the maximum tension and cause various types of cracks in a slab. Yang Jinzhi [48] analyzed the deformation and crack generation mechanism of concrete pavement surface under the coupling effect of temperature and cyclic load by using finite element software, and obtained the relationship among temperature, load, and pavement displacement. The failure law of pavement under the coupling effect of temperature and the cyclic load was obtained. Some valuable conclusions for concrete pavement construction were obtained. Hu Changbin et al. [49] studied the temperature field and temperature stress of cement concrete pavement in hot and humid areas, and concluded that “with the periodic change of external environmental meteorological conditions, there were many typical distribution shapes of pavement temperature gradient and they changed with the external environment”. Combined with relevant research experience and results [6,50,51,52], the deformation and cracking of pavement concrete during the service period are related to the times of wheel load action and wheel load impact effect, and closely related to the temperature and humidity changes in the actual use environment. Especially for areas where the temperature and humidity changed dramatically in day and night or season, it was very important to study the deformation and cracking behavior and meso mechanical mechanism of concrete pavement under the coupling action of moisture-heat-force.

## 5. Research on Deformation and Failure of Concrete (Except for Airport Pavement and Highway Pavement) under the Action of Moisture-Heat-Force

The research on the hygro-thermal-mechanical coupling action analysis was previously in the field of rock engineering and had always been a research hotspot [53,54]. The moisture-heat-force multi-field coupling research in the field of rock engineering was basically about the deterioration of concrete performance under high temperature or fire [55,56,57,58,59,60,61]. Schrefler et al. [55,56] put forward the mathematical and numerical model of the nonlinear performance of porous multiphase concrete according to the characteristics of porous multiphase concrete and the principle of thermodynamic equilibrium. The early performance of self-drying and high-temperature deformation of concrete was simulated and analyzed. A moisture-heat-force coupling model of concrete at high temperature was proposed, its numerical simulation was realized by a finite element method, and the performance of high-performance concrete walls and columns in the fire was studied. Through moving boundary problems, Beneš and Štefan [59] put forward a mathematical model of hygro-thermal-mechanical analysis for a high-temperature burst of the concrete wall. The thermal stress effect and pore pressure change of burst were considered, and the validity was verified. S. Grasberger and G. Meschke [62] established a 3D coupled thermo-hygro-mechanical model for concrete accounting for moisture and heat transport, cracking and irreversible deformations, and the various interactions between these processes. The effects of drying shrinkage, non-isothermal transmission, and cracking on the drying process of concrete were studied. Bangert et al. [63] considered the mechanical damage and the interaction of moisture heat transfer and established a concrete hygro-thermal-mechanical coupling model based on the theoretical mechanism analysis and experimental results. A two-dimensional simulation of a concrete slab under the condition of moisture thermal-mechanical coupling was carried out. D. Gawin et al. [64,65] regarded concrete as a multiphase porous material and proposed a mathematical model for analyzing the hygro-thermal behavior of high-temperature concrete. Jaroslav Kruis et al. [66] studied and analyzed the effective computer implementation of the coupled analysis of prestressed concrete nuclear reactor based on the hygro-thermal-mechanical coupled analysis. The hydro-thermo-mechanical analysis of reactor vessels based on the finite element method was a very difficult task because of its complexity and numerous unknowns. This contribution involved efficient computer implementation of coupling analysis and was also devoted to the implementation of domain decomposition methods that could utilize parallel computers. Parallel processing could achieve very good acceleration and solve large problems in an acceptable time. The proposed strategy is demonstrated in the coupling analysis of existing reactor vessels. C.T. Davie et al. [67] analyzed the sensitivity of typical prestressed pressure vessels through the hygro-thermal-mechanical fully coupled model of concrete. The results showed that changes to operating procedures only led to minor changes in the behavior of the structure throughout its life cycle but the unplanned thermal excursions could have a greater impact on the concrete structure. Li Rongtao and Li Xikun [57,58] studied the failure process of concrete at high temperature, the constitutive relationship of concrete at high temperature, the chemical hygro-thermal-mechanical coupling process, and the numerical calculation method. Li Zhongyou and Liu Yuanxue [68] studied the evolution process of mechanical damage, which started from the characteristics of energy dissipation in the process of material deformation (failure) and based on the mixture theory. Considering the thermal damage caused by high temperature, the heat-water-force coupling damage model of concrete under high temperature was established. Liu Jiaping et al. [69] studied the early cracking of sidewall concrete. A multi-field (hygro–thermo-chemo-mechanical) coupling model based on Fourier’s law, Fick’s law, and mass and energy balance equations, was adopted to describe the cement hydration, temperature, and humidity evolution for early-age sidewall concrete. Gasch et al. [70,71] established the hygro-thermal-mechanical coupling model of hardened concrete based on microprestress-consolidation theory, which took into account the factors such as age, creep, shrinkage, thermal expansion, and cracking under the condition of varying temperature and humidity, which had inspirations on this project.

## 6. Numerical Simulation Method of Hygro-Thermal-Mechanical Coupling Deformation of Concrete

In the research of the numerical simulation method of hygro-thermal-mechanical coupling deformation of concrete, there are four main numerical simulation methods:

### 6.1. Heat and Mass Transfer in Porous Media of Phenomenological Thermodynamic Method

This method does not involve the internal heat and mass transfer mechanism and specific processes of the porous media, but only considers the relationship and cross effect between various flows and forces of heat and mass transfer. A phenomenological equation can be obtained to describe various flows. Phenomenological methods are difficult to apply in practice due to the influence of test conditions and many material parameters. Due to the lack of effective means and methods for testing moisture distribution, the moisture migration rate is slower than the thermal migration rate. In the process of solving various models, the values of some coefficients are basically based on experimental or empirical data, which affects the versatility of this method.

### 6.2. Numerical Analysis Method Based on Luikov’s Coupled Heat and Mass Transfer Equation

This method considers the heat absorption or exotherm in the process of heat and mass transfer, and establishes a partial differential equation system based on the principles of mass conservation and energy conservation in the process of moisture migration. The method of numerical analysis is used to solve the equations to obtain the analytical solutions of the heat, humidity fields, and their dynamic changes [72,73,74,75]. This method is also mainly used for the analysis of heat and mass transfer in porous media. With the development of computer technology, the difficulty of implementing numerical analysis methods is greatly reduced. Concrete is also a typical porous medium [5,6] and the above method can be considered for application in concrete. In the process of solving the temperature and humidity coupling equations, there are different solutions; for example, Bouddour [76] et al. used the asymptotic continuous method of periodic structure to study the heat and humidity transfer in the evaporation and condensation process. However, the research methods involved material did not absorb wet steam, and the hygroscopicity of the material had an effect on the heat transfer process. Lobo and Mikhailov [77,78] used the classic integral transformation method to solve the heat and mass transfer problem in porous media. Because of its complex characteristics and difficult calculation, it could not accurately reflect the distribution of temperature and humidity to a large extent, and can not get the correct results. Chang [79] et al. used decoupling technology to solve the coupled equations of the heat and mass transfer process; however, this appeared powerless when the governing equations and boundary conditions were coupled at the same time. Cheroto [80] et al. used an improved lumped system analysis method to find the approximate solution of the coupling equation. Although it avoided the problems encountered in the calculation of complex eigenvalues and Chang’s decoupling technology, its accuracy was not enough. It can not truly reflect the temperature and humidity distribution in porous media.

### 6.3. Finite Element Analysis Method

The finite element method is the most widely used numerical calculation method in scientific research and has become the main tool for solving scientific and technological problems. At present, many commercial finite element softwares have been developed, such as ANSYS, MSC, ADINA, ABAQUS, etc. In the finite element analysis method, the model should be established first, and then solved by self-programming [57,58,63,81], ANSYS and ABAQUS software are used more in pratice [82,83,84,85]. Bernard et al. [82] proposed a three-dimensional multi-scale simulation method for the mechanical properties of cement-based materials. Firstly, the CEMHYD3D model was used to generate ideal three-dimensional representative volume elements of cement-based materials at different scales, and then the mechanical behavior was calculated by using the finite element analysis software ABAQUS. Li Zhaoxia et al. [83,84,85] studied the damage analysis and state assessment of long-span structures. A series of multi-scale modeling processes, multi-scale damage, and its homogenization algorithms in the structural multi-scale simulation were studied based on the nonlinear finite element software ABAQUS.

### 6.4. Application of Multiphysics Coupling Analysis Software

The analysis method of COMSOL multi-physical field coupling calculation software [6,70,86] was used to model and simulate engineering problems based on partial differential equations. This method was easy for a couple of different physical fields for simulation. In the author’s previous research, the analytical calculation of the temperature and humidity field based on the Luikov humidity–heat coupled transfer equation had been studied carefully. The analytical calculation of temperature and humidity field and finite element analysis of coupled deformation were realized by mixed programming based on the principle of heat and moisture transfer in porous media. The above research can simulate the moisture heat coupling deformation behavior of some concrete within a certain engineering precision range. The differential equation of this model had considered the mutual influence between temperature and humidity, which reflected the coupling of humidity and heat, but it was more difficult to solve. Moreover, through the Laplace transform and the transfer function method, the calculation requirements were stricter, and the selection of the transfer function was more difficult, especially for the study of the internal temperature and humidity distribution of the concrete under the change of environmental temperature and humidity conditions. The author published an academic paper [6] in 2013. Based on the theory of moisture and heat transfer in porous media and microprestress consolidation theory, the coupled model of concrete moisture-heat-force was established, and the numerical solution of the coupling was realized by using COMSOL multi-physical field numerical simulation software and proposed the concrete moisture expansion coefficient. Gasch et al. [70] published a paper in 2016, using COMSOL software to carry out coupled finite element simulation analysis of strong form partial differential equation, which has important reference value for this research. 

## 7. Conclusions and Perspective

The studies of the coupled hygro-thermo-mechanical behavior of concrete pavement have been reviewed in this paper. The various aspects on the deformation and failure of concrete pavement under coupling action of moisture, temperature and wheel load presented in this paper could be summarized and concluded as:

1. Concrete is a typical porous material, moisture and heat transfer theory had been obtained enough data to simulate the hygro-thermo-mechanical properties of concrete, and the relationship between moisture and heat is very clear.

2. There has been some research on the moisture-heat-mechanical coupling of concrete materials, but most of the research focused on the performance of concrete in fire or high temperature. There are few studies on the moisture-heat-mechanical coupling deformation of concrete under normal service conditions.

3. In the related research on deformation and failure of airport pavement or cement concrete pavement, many researchers mainly focused on the influence of temperature, fatigue load and freeze-thaw, and temperature and humidity changes, which lead to the shrinkage and cracking of pavement concrete. There are few kinds of literature considering the coupling action of moisture, temperature, and wheel load at the same time.

4. Concrete pavement is subjected to hygro-thermal-mechanical coupled action in service, which has the characteristics of a similar period and its possible fatigue effect. In the existing moisture-heat-mechanical coupling research, the research on “force” is mostly the load of a certain stress level continuously applied, but not enough attention is paid to the periodicity of wheel load and the change of temperature and humidity.

5. COMSOL software has certain advantages for solving the coupled hygro-thermal-mechanical of concrete.

In the later research, the deformation and failure mechanism of pavement concrete under the coupling action of moisture, temperature, and wheel load need to be studied.

## Figures and Tables

**Figure 1 materials-13-05530-f001:**
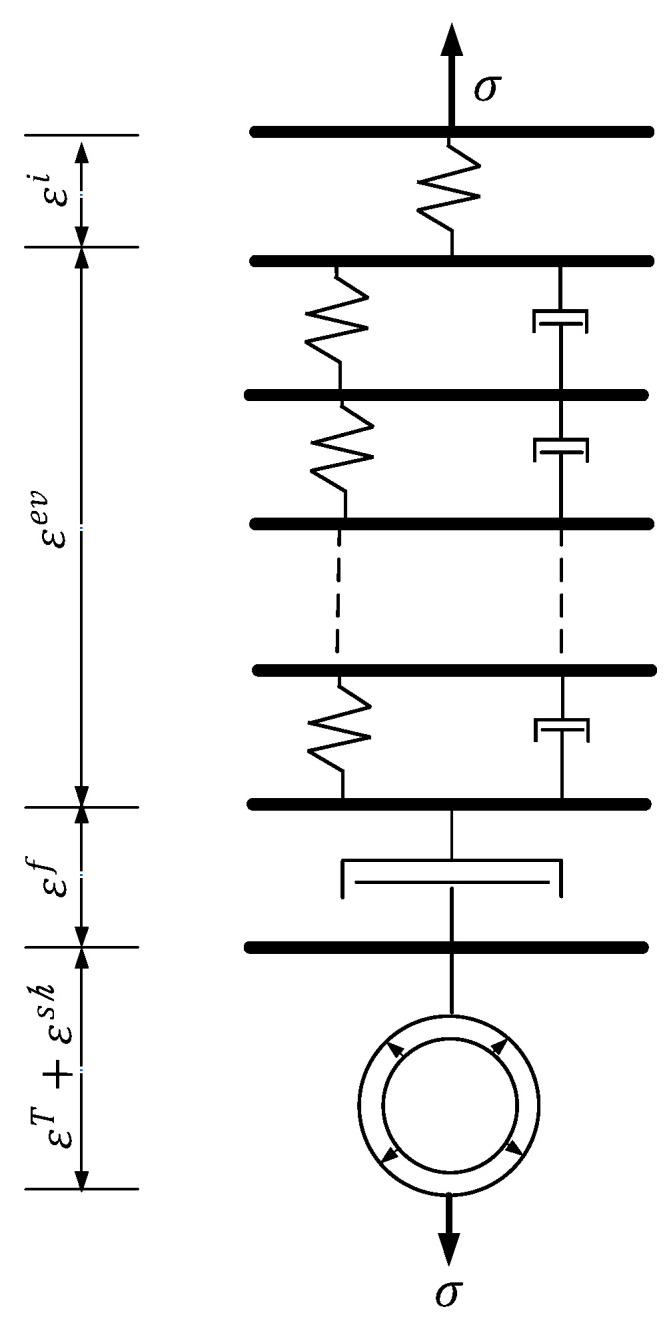
Rheological model.

**Figure 2 materials-13-05530-f002:**
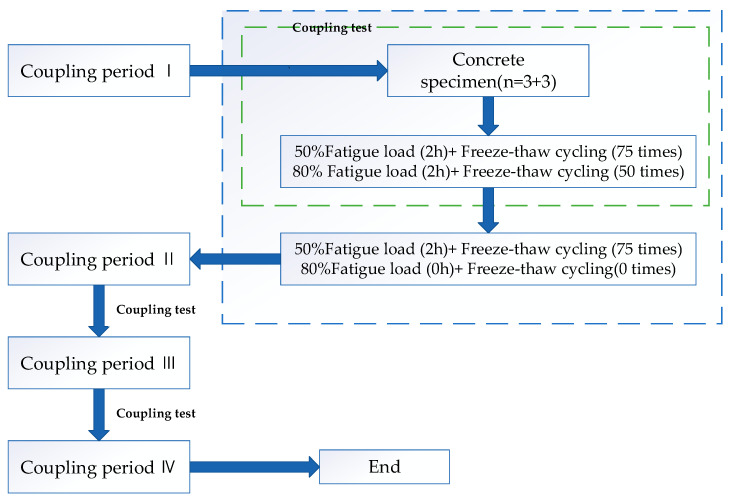
Flow chart of the double-field coupling experiment [30].

**Figure 3 materials-13-05530-f003:**
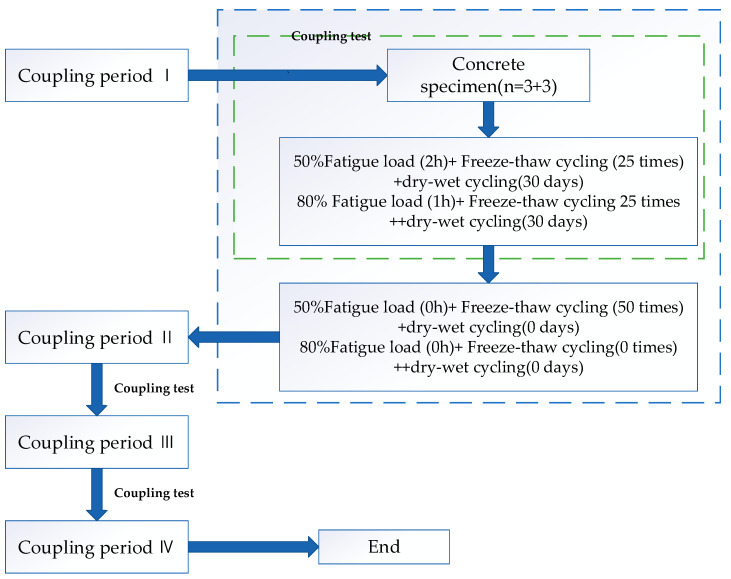
Flow chart of the triple-field coupling experiment [30].

**Figure 4 materials-13-05530-f004:**
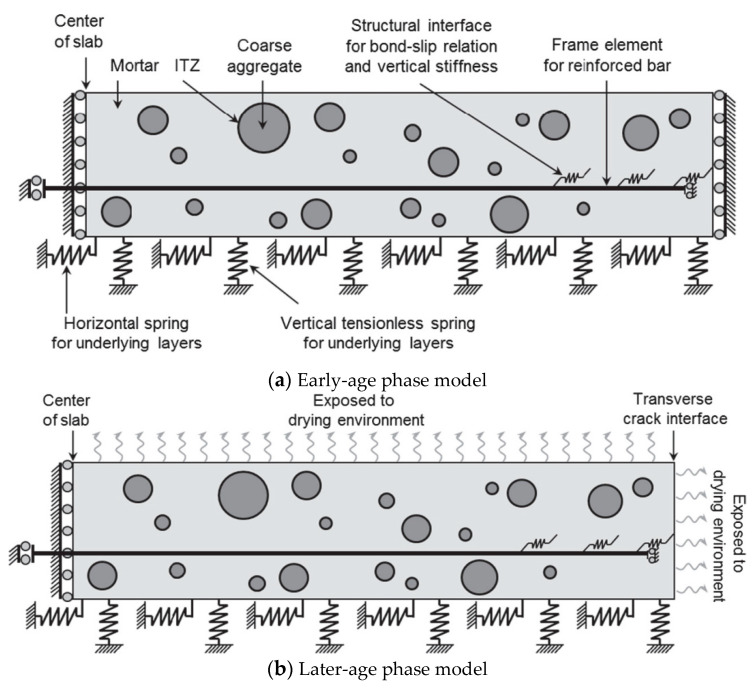
Early-age and Later-age phase model [29].

**Table 1 materials-13-05530-t001:** Research progress abroad.

Number of Cycles	Author	Viewpoints
1	Shadravan. Shideh [24]	Study on the behavior of concrete slabs on the ground in controlled moisture and temperature environment.
2	Maekawa. Koichi [25]	Structural creep deformations were reproduced by using the multi-scale coupled thermo-hygro-mechanical modeling.
3	Ye. Dan [26]	Moisture capacity was induced into the temperature and moisture analysis for curing concrete to solve the coupled and nonlinear heat transfer and moisture transport problems in early-age concrete.
4	Hiller. Jacob E. [27]	A simplified method termed NOLA(Nonlinear Area) and a mechanistic-based rigid pavement analysis program called RadiCAL was proposed to determine fatigue damage levels and critical cracking locations.
5	Simonova. Anna [28]	The scope of the research is to analyze the impact of ambient temperature variations in water, and the thermal state of the subgrade on different types of road structures.
6	Seongcheol. Choi [29]	Mesoscale analysis of continuously reinforced concrete pavement behavior subjected to temperature and moisture variations.

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
