# Peer review of "Advances in the Deformation and Failure of Concrete Pavement under Coupling Action of Moisture, Temperature, and Wheel Load"

_materials, 2020, doi:10.3390/ma13235530_

Round 1

Reviewer 1 Report

Title is not clear and should consider rewording:

Advance in deformation and failure of pavement concrete under coupling action of moisture, temperature and wheel load  

should be:

Advances in the deformation and failure of concrete pavement under coupling action of moisture, temperature and wheel load

Overall Assessment

This paper is meant to be an overview; however, the topic is not stretched out enough with many subsections to capture all the work done in this area. This topic is wider than the 3 sectionals (2, 3 and 4) devoted to reviewing the work covered in the field. A review should be multi-structured with subsections that address different parts of the available work.

In my view this review still needs a lot more in terms of literature.

Abstract

Abstract not well composed and should be rewritten to reflect the content of purpose, procedures and principal findings.

Line 22– 31:

It is necessary to study the deformation and failure behavior of pavement concrete and its micro-mechanical mechanism …  porous media, micro prestress-solidification, and damage mechanics.

The sentence is too long to follow and should be broken down into many sentences.

Introduction

Too many long sentences that are difficult to follow. These should be broken down into shorter ones. Examples are:

Line 61 – 66:  

The main reason why 61 the deformation and cracking of airport runway, … and wheel load are not well understood.

Line 70-74

The micromechanical mechanism of … service performance and service life of the structure.

line 103 – 109

At present, there are few literatures on the meso-mechanical mechanism of deformation … deformation and 108 failure under the coupled action of moisture-heat-wheel load

Avoid the use of “Foreign” and  “domestic”.   This paper is intended for an international audience.

Should define “CRCP” on its first mention.  

Conclusion and Perspective

Again, many long sentences that makes narrative difficult to follow. The following should be broken into shorter sentences.

Line 292 - 297

In the previous research, the analytical calculation of temperature … deformation behavior of some concrete within a certain engineering precision range.

Line 299 – 303

Moreover, through 299 Laplace transformation and transfer … it is impossible to complete the task. The author published 303 an academic paper [29] in 2013.

Line 315 – 318.

Based on the principle of moisture heat transfer in porous media, … damage effect and the internal damage evolution of pavement concrete are studied

Author Response

Dear Editors and Reviewers:

We very much appreciate your insightful comments and critiques. These comments have been very helpful to us when revising the paper. We have carried out a comprehensive revision of the manuscript on these comments. The revisions are addressed point by point below and are marked in red in the main manuscript.

Reviewer 2 Report

The aspects discussed in the reviewed paper include deformation and failure of airport pavement or cement concrete pavement damage of concrete under the coupling hydro-thermo-mechanical  action. In reviewer opinion the paper is not clear enough to present the problem. There are some comments:

  1. What is the aim of the paper? If there is the state-of-the-art as it was mentioned in the abstract the paper should be reorganized. The aim should be clearly presented in the introduction.
  2. Many informations and problems that are which were mentioned in the article are repeated.
  3. There are abbreviations that should be explained, for ex. CRCP.
  4. There is no analysis in the paper. 
  5. The conclusions are not really conclusions.
  6. The paper ends rapidly in line 322 and it seems to be unfinished.

Author Response

(The authors gave the same response as above.)

Reviewer 3 Report

This Review is on the deformation and failure of pavement concrete under coupling action of moisture, temperature and wheel load.

In the Review, neither modeled nor experimental, quantitative data are available.
Authors' own contributions to the literature with this review, and differences from the published review reports are not clear.
My suggestion is that the authors should add the quantitative data comparisons including the basic knowledge.

Authors have not discussed the latest reports. Articles published in the last 2 years have not been evaluated.There are more than 200 related references published in 2020. None of them are discussed in the article.

I would suggest revision this review and accept it after adding quantitative data comparisons from theoretical or experimental studies and the most recently published reference discussions.

Details:
In the Introduction section, there are missing references for the sentences such as;
In Line 36: "More than 80% of the airport pavement in China adopts cement concrete.
In Line 41: "At present, there are more researches on temperature cracks."
In Line 103: "At present, there are few literatures on the meso-mechanical mechanism of deformation and failure of pavement concrete structures...."

There are sentences that need revising such as;
In Line 77: "....material properties and properties[1]."
In Line 81: "...micro and micro composition..."
In Line 115: "Choi[8]and others..."
In Line 123: "Dhawale A W [9] thinks..."
In Line 195: "...of concrete (not for airport pavement..." you may write "(except for..."
In Line 238: "...for this project."
In Line 248: "...software are more used..."
In Line 257: "ABAQUS. Li Zhaoxia et al. [55-57] in order to meet the needs of finite..."

In Line 121: CRCP explanation should be given at the first mentioned line.
In Line 132: First Author of the Reference [12] is not Ye. D..
In Line 136: What does "Many foreign researchers" mean?
In Line 309: "...which has important reference value."

There are typing errors such as;
In Line 31: "...mechanics. the..."
In Line 127: "...More..."
In Line 132: "Ye [11, 12]studied.."
In Table 1: some words should be capitalized like "author"
In Line 328: "...nos. 51..."

Author Response

(The authors gave the same response as above.)

Reviewer 4 Report

This paper presents a state of the art review on the deformation and failure of pavement
concrete under coupling action of moisture, temperature and wheel load

The main topic is interesting but the paper needs major revision.

However, the remarks below are provided in order to improve the paper: (Both editorial and technical comments are given in the order they appear in the text.)

- Pag 3, line 115-> Please change "Choi[8]and others " into "Choi et al. [8]".

- pag 3, line 121-> Please define the acronym CRCP.

- line 127-> A dot is missing between "stress" and "More"

- Many blanks are missing between the author and the reference number under bracket []. See for example:
Lie 128-> Tomas Yu[10]
line 132-> Ye [11, 12]studied

- line 132-> Please clarify the difference between "warping stress" and "warpage stress" used at line 125.

- line 136-> The expression "Many foreign researchers" in an international journal seems inappropriate. Please reconsider the this sentence.

- line 147-> The expression "Many domestic researchers" in an international journal seems inappropriate. Please change it into "Many Chinese researchers" .

- Please cite every Figure and every table in the text. Table 1 and Figure 3 are not cited.

- The weakest point of the review is that there is not accurate or clear description of the available models. Thus, the reader at the end has not a thorough view of the possible approach to the problem.

- I would suggest to highlight what are the experimental studies on the problem.

- Finally it is very important to point out why is important to consider the coupling action and not each singular one. It would be great to have a numerical example to quantitatevely explain to the reader the need of this approach.

Author Response

(The authors gave the same response as above.)

Round 2

Reviewer 1 Report

The authors have addressed most of the questions raised.  I would expect that the editors would tidy up the structure prior to publishing.

Author Response

Dear  Reviewers:

We would like to thank you for your insightful comments and comments, which have been of great help to us in revising the paper. We have made a comprehensive revision of these comments and promised to conduct extensive editing of the English language and style before publishing.

Kind regards

Dr. dong

Reviewer 2 Report

Thank you for all the improvements of the paper. 

Author Response

(The authors gave the same response as above.)

Reviewer 4 Report

This paper presents a state of the art review on the deformation and failure of pavement concrete under coupling action of moisture, temperature and wheel load

I appreciate the work done by the authors to improve the paper.

I can assess that the English level is still poor. For example:

line 369-570-> It is found that COMSOL software has a good effect on the simulation of concrete deformation under the hygro-thermo-mechanical action.

Please perform a professional proofreading with an English mothertongue speaker.

Author Response

(The authors gave the same response as above.)
